# Ideal Life’s Simple 7 Score Relates to Macrovascular Structure and Function in the Healthy Population

**DOI:** 10.3390/nu14173616

**Published:** 2022-09-01

**Authors:** Gilles Nève, Jonathan Wagner, Raphael Knaier, Denis Infanger, Christopher Klenk, Justin Carrard, Timo Hinrichs, Henner Hanssen, Arno Schmidt-Trucksäss, Karsten Königstein

**Affiliations:** 1Division of Sports and Exercise Medicine, Department of Sport, Exercise and Health, University of Basel, Birsstrasse 320B, CH-4052 Basel, Switzerland; 2Institute for Diagnostic and Interventional Radiology, Klinikum Rechts der Isar, School of Medicine, Technical University of Munich, 81675 Munich, Germany; 3Clinic for Children and Adolescent Medicine, Städtisches Klinikum Karlsruhe, 76133 Karlsruhe, Germany

**Keywords:** Life’s Simple 7, carotid intima–media thickness, flow-mediated dilation, pulse wave velocity, lifestyle, physical activity

## Abstract

Background: Cardiovascular health scores, such as Life’s Simple 7 from the American Heart Association, and the assessment of arterial properties are independently used to determine cardiovascular risk. However, evidence of their association remains scarce, especially in healthy, middle-aged to older populations. Methods: A healthy sample of the Swiss population aged 50–91 years as part of the COmPLETE cohort study was included. Carotid intima–media thickness (cIMT), carotid lumen diameter (cLD), carotid distensibility coefficient (DC), flow-mediated dilation (FMD), and brachial–ankle pulse wave velocity (baPWV) were used to determine arterial properties. The Life’s Simple 7 cardiovascular health score was calculated using seven categories (body-mass index, cholesterol, systolic blood pressure, hemoglobin A1c, smoking status, physical activity, and diet). In accordance with the American Heart Association, for each category, two points were given for an ideal health metric level, intermediate scores one point, and poor scores zero points. Intermediate and ideal health scores corresponded to a total of 5–9 and 10–14 points, respectively. Results: A total of 280 participants (50.7% male) were included. After adjusting for age and sex, an ideal health score was associated with lower cIMT (−0.038 mm, 95% CI: −0.069 mm–−0.007 mm, *p* = 0.017), lower cLD (−0.28 mm, 95% CI: −0.46 mm–−0.11 mm, *p* = 0.002), and lower baPWV (−0.05 m/s, 95% CI: −0.08 m/s–−0.02 m/s, *p* = 0.003). No differences were found for FMD and DC. Conclusions: Even in a healthy sample of middle-aged and older adults, individuals with an ideal cardiovascular health score showed more favorable biomarkers of vascular aging than those with an intermediate score. This stresses the relevance of promoting an optimal lifestyle, even among the healthy population.

## 1. Introduction

Age is the strongest independent risk factor for cardiovascular morbidity and mortality [1]. Maintaining good health until old age plays an important role [2]. Pursuing a healthy lifestyle is a key determinant in delaying the onset of cardiovascular disease. As such, the American Heart Association introduced Life’s Simple 7, consisting of 7 modifiable and easily applicable lifestyle metrics [3]. Adherence to Life’s Simple 7 has been linked to a longer and healthier life, but in the USA, the percentage of people pursuing a healthy lifestyle has steadily been on the decline [4]. In Switzerland, recent data suggest that physical activity is on the rise [5], and more people are pursuing a healthy diet [6]. However, improvements need to be made for tobacco smoking [7] and alcohol consumption [8].

Despite offering a simple approach to implementing lifestyle changes in a patient’s life, adherence to Life’s Simple 7 has been shown to influence vascular aging and cardiovascular morbidity and mortality. The Atherosclerosis Risk in Communities (ARIC) study demonstrated a reduced incidence of arterial cardiovascular diseases [9] and venous thromboembolism [10], and lesser arterial stiffness at higher age, in those participants that adhered to Life’s Simple 7 [11]. Other studies demonstrated a high effect of Life’s Simple 7, especially in a population with a generally low proportion of individuals with an optimal lifestyle, on carotid intima–media thickness (cIMT) [12,13,14] and coronary artery calcification [13]. However, to the best of our knowledge, there has been no study assessing the association between ideal cardiovascular health metrics and clinical biomarkers of vascular health in a healthy middle-aged to elderly population.

This is somewhat surprising, as extensive evidence on assessing cardiovascular risk using arterial structure and function has been published. cIMT is widely used as a surrogate for cardiovascular risk as it reflects subclinical atherosclerosis [15]. Recently, pooled analyses of four cohort studies in Europe suggested that carotid lumen diameter (cLD) might improve the prediction of stroke, cardiovascular disease, and mortality beyond cIMT [16]. While men have higher cIMT and cLD than women, previous studies have suggested that progression is similar for both sexes [17,18].

In addition, brachial flow-mediated dilation (FMD) has emerged as a valid noninvasive tool to detect early changes in cardiovascular functionality, and thus to predict future cardiovascular events [19]. It has previously been shown that FMD decreases with age [20] and can be used to improve the classification of aging participants with low, intermediate, and high cardiovascular risk [19,21].

Further, arterial stiffness measured by brachial–ankle pulse wave velocity (baPWV) is an independent predictor of cardiovascular events, even in healthy adults [22]. Arterial stiffening, a well-recognized indicator of early vascular aging, results from arterial wall remodeling driven by lifestyle factors, such as diet and physical activity [23].

In the general population, Life’s Simple 7 and various ultrasound-based assessment methods of arterial structure and function are independently known to be influenced by age and lifestyle. Both have been used separately to assess cardiovascular health, but evidence of their association is scarce, especially in healthy middle-aged to elderly populations. Therefore, this study aims to assess the association between the Life’s Simple 7 cardiovascular health score and clinical biomarkers of structure and function of the main arteries in healthy community-dwelling participants.

## 2. Methods

### 2.1. Population and Recruitment

The study consisted of participants between 50 and 91 years of age who were part of the COmPLETE Health study at the University of Basel and were tested between January 2018 and June 2019. The study was designed to identify healthy individuals who may serve as perfect examples to characterize healthy aging in the Swiss population. The full study protocol has been published previously [24]. Briefly, all participants were invited via letters sent to various districts of the city of Basel, Switzerland, and its suburbs. To be included in the COmPLETE Health study, all participants were to be non-smokers for at least 10 years, have blood pressure values < 160/100 mmHg, be free of any cardiovascular disease, and have a BMI < 30 kg/m^2^. Exclusion criteria included drug or alcohol abuse, pregnancy, breastfeeding, history of cancer, or the inability to follow the study procedures. For the present study, only participants aged 50+ years were included, as we did not measure carotid properties in those who were younger than 50 years old. Participants were asked to refrain from any vigorous physical activity 24 h before the measurement. Further, abstaining from alcohol consumption or caffeine on the day of the examination, as well as fasting for three hours, was requested. All the vascular screenings were performed in a supine position after the participants rested for at least 10 min in a dimly lighted, quiet room with ambient temperature. All participants were asked to wear noise-canceling headphones. At the start of the single visit, written informed consent was obtained from all participants before undergoing any study procedures. The study was approved by the Ethics Committee of Northwestern and Central Switzerland (EKNZ 2017-01451) and complied with the Declaration of Helsinki.

### 2.2. Carotid Properties

Carotid intima–media thickness (cIMT) measurement was performed using a standardized ultrasound scan protocol using the Fukuda UF 760 ultrasound scanner (Fukuda Denshi, Toyko, Japan) with an FUT-LA385-12P (8–13 MHz) transducer (Fukuda Denshi, Tokyo, Japan) [25]. The measurements were limited to the right common carotid artery to determine cIMT and carotid stiffness. Participants were asked to stay supine for 10 min with the head rotated at 45° to facilitate measurement. The ultrasound was performed along the common carotid artery and proximal to the carotid bifurcation over the course of two to six heart cycles. Software-based electrocardiogram-gated real-time quality control with automatic validation was used (Fukuda Denshi, Tokyo, Japan). This was based on a previously developed detection algorithm to assure accurate wall detection [26]. Inner carotid lumen diameter (cLD) and carotid distensibility coefficient (DC) were measured simultaneously. DC was calculated as $\frac{\left(2\;\times \;delta\;cLD\;\times \;diastolic\;cLD\right){+(delta\;cLD)}^{2}}{{\left(PP\;\times \;diastolic\;cLD\right)}^{2}}$, whereas delta cLD equated to systolic cLD minus diastolic cLD, and pulse pressure (PP) equated to systolic blood pressure minus diastolic blood pressure. In addition, blood pressure was measured oscillometrically on both arms (OMRON 705IT, OMRON Healthcare, Kyoto, Japan).

### 2.3. Pulse Wave Velocity

PWV was measured as brachial–ankle PWV (baPWV) using a noninvasive vascular screening device (VaSera VS-1500 N; Fukuda Denshi, Tokyo, Japan). Blood pressure cuffs were placed around the left ankle and left upper arm, and ECG recording was performed simultaneously. Using the foot-to-foot method for noninvasive measurement of baPWV, the pulse wave time from the heart to the ankle was calculated. The estimation of baPWV was done using a height-based formula by the VSS-30 software (Fukuda Denshi, Tokyo, Japan) [27]. The average of two measurements was taken to determine blood pressure and baPWV. If the difference between the two measurements was >10 mmHg for systolic blood pressure, a third measurement was conducted. The average of the lowest two values was then used.

### 2.4. Endothelial Function

FMD was measured semiautomatically using a high-resolution ECG-guided B-mode ultrasound system (UNEX EF 38G, UNEX Corp., Nagoya, Japan) according to current guidelines [28]. A 10-MHz H-shaped probe was used to generate short-axis and long-axis images of the right brachial artery and continuous automatic correction of the probe position during the whole measurement period. During the procedure, the arm was abducted at a 90° angle, and the insonation angle of the probe was fixed between 60 and 70°. The proximal edge of the cuff was placed 1–2 cm proximal to the cubital fossa and 5–10 cm distal to the probe. Pre-cuff-inflation diameter was measured for 10 s immediately before the 5 min occlusion period. During the occlusion period, cuff pressure exceeded systolic blood pressure by 50 mmHg. During the last 60 s, pre-cuff-deflation diameter was measured. Post-deflation diameter and blood flow velocity were continuously measured during the 3 min post-deflation period, and end-diastolic values (ECG-guided) were recorded for subsequent data analysis. After data collection, blinded semiautomatic quality control of all measurements was conducted using the UNEX EF 38G 1.0.14 software (UNEX Corp., Nagoya, Japan).

### 2.5. Physical Activity

Physical activity was measured objectively via a wrist-worn triaxial accelerometer at a frequency of 50 Hz (GeneActive Activinsights Ltd., Kimbolton, UK) for 14 days. All participants were instructed to wear the accelerometer on the nondominant wrist during the measurement period and return it on the 15th day. The data were exported via the GENEActiv software v3.2 (GeneActiv Activinsights Ltd., Cambridgeshire, UK) and analyzed via the open-source Excel macro file “General Physical Activity” version 2 (Activinsights Ltd., Cambridgeshire, UK) [29]. The cutoffs for physical activity intensity were defined as 1.5–3.99 MET for low physical activity, 4.0–6.99 MET for moderate physical activity, and ≥7 MET for vigorous physical activity. Cutoffs for valid days were defined as follows: (1) at least 10 h of wear time over one day (midnight to midnight), (2) less than 18 h of low physical activity, (3) less than 8 h of moderate physical activity, and (4) less than 2.5 h of vigorous physical activity. Participants were only included in the analyses if at least five weekdays and two weekend days were valid [30]. For all valid days, the number of minutes of low, moderate, and vigorous physical activity was averaged.

### 2.6. Nutrition

Nutrition status was calculated using a validated dietary assessment tool [31]. Dietary assessment tools have been useful in estimating dietary habits, such as the amount of fruit and vegetables consumed. This dietary assessment tool displays the food pyramid of the Swiss Society for Nutrition (version 2005-2011). A portion size equivalent for various food items of the respective category is pictured in the middle, as well as five mealtimes (breakfast, snack #1, lunch, snack #2, and dinner). On the right side, there is a column for the sum of the five mealtimes. All participants were asked to fill out the dietary assessment tool for a typical day (e.g., normal workday, illness-free) and to write the number of portions per food item per mealtime. Subsequently, the kilocalories of the different food groups were calculated via a portion size equivalent.

### 2.7. Other Measurements

Body height and mass were measured to the nearest 0.5 cm and 0.1 kg. Body-mass index was calculated as weight in kg divided by the squared value of height in meters. Smoking status was assessed before the visit via telephone interview. For hemoglobin A1c (HbA1c) and cholesterol, blood samples were collected by standard laboratory procedures with the Cobas analyzer (Cobas 8000; Roche Diagnostics, Basel, Switzerland).

#### Cardiovascular Health Score

As a surrogate for ideal cardiovascular health, we used the Life’s Simple 7 score, introduced by the American Heart Association in 2010 [3]. The health score incorporates four modifiable behavioral metrics, namely physical activity, diet, smoking status, and BMI, as well as three biological metrics, namely systolic blood pressure, HbA1c, and cholesterol. For each health metric used for the score, a study participant could either score 0 (poor), 1 (intermediate), or 2 (ideal) points. Therefore, the health score ranged from 0 (poor) to 14 (ideal). The limits of each variable used in the health score were the same as those presented by the American Heart Association [3]. Scoring 0–4 points was considered poor, 5–9 intermediate, and 10–14 ideal.

The dietary component of Life’s Simple 7 consists of 4 goals, which are ≥400 g of vegetables per day, ≥2 servings of fish per week (=30 g of fish per day), ≥14 g of fiber per 1000 kcal, and <10 energy percentage of daily intake of saturated fatty acids. The dietary assessment tool used in this study does not allow the assessment of all four components. Therefore, we used the daily servings of fruits and vegetables as a proxy for diet quality. Poor diet was considered ≤3 servings/day, 3–4.9 as intermediate, and ≥5 as ideal. In addition, we did not measure fasting blood glucose but HbA1c. We used the cutoff values for poor, intermediate, and ideal scores that were suggested by Ford et al. (2012), which determined ≥6.5% as poor, 5.7% to <6.5% as intermediate, and <5.7% as ideal [32].

### 2.8. Statistical Analyses

Means and standard deviations were calculated for the descriptive section of our dataset. We performed independent *t*-tests to calculate the differences between the sexes.

We used multiple linear regression models to compare the outcomes (cIMT, cLD, DC, FMD, and baPWV) between individuals with an ideal and an intermediate health score. All models were adjusted for sex and age. In addition, cLD was adjusted for height [33]. We included age using restricted cubic splines with four knots placed at specific data percentiles to account for potential nonlinear associations [34] and an interaction between age and sex. Due to missing data, complete-case analyses would have resulted in the loss of 1% of observations for cIMT and 23%, 23%, 24%, and 0% for cLD, DC, FMD, and baPWV. Hence, we handled missing data using multiple imputation [35,36]. Specifically, we used predictive mean matching to impute 50 datasets using all outcome variables, age, sex, health score, and height for prediction. Adequacy of model fits was checked using Q-Q-plots of the residuals and plots of fitted values versus residuals. As the residuals of the models for DC and baPWV showed a marked deviation from normality, we log-transformed those outcomes, after which the model assumptions were satisfied. For the log-transformed effects, we presented the exponentiated estimates and confidence intervals, which represent the ratio of the geometric means of the ideal outcomes to the intermediate health score.

Significance was set at *p* < 0.05 in all tests; all tests were two-sided. All analyses were performed using SPSS version 26.0 for Windows (SPSS Inc., Chicago, IL, USA) and R version 4.2.0 for Windows (R Foundation for Statistical Computing, Vienna, Austria).

## 3. Results

Data from 280 participants (50.7% male) aged 50–91 years were included in the analyses (Figure 1). The participants’ characteristics are described in Table 1. In total, 125 participants (50.4% male) had an ideal cardiovascular health score, 155 participants (51.0% male) had an intermediate cardiovascular health score, and none of the participants had a poor cardiovascular health score.

All analyses of the arterial properties were adjusted for sex and age. As shown in Table 2, those with an ideal health score had lower cIMT than those with an intermediate score (−0.038 mm, 95% CI: −0.069 mm–−0.007 mm, *p* = 0.017). Similarly, those with an ideal health score had lower cLD than those with an intermediate score (−0.28 mm, 95% CI: −0.46 mm–−0.11 mm, *p* = 0.002). Those with an ideal health score had a 5% lower geometric mean of baPWV (geometric mean ratio: 0.95) compared to those with an intermediate health score (95% CI: 0.92–0.98, *p* = 0.003). Figure 2 shows the absolute means of baPWV by cardiovascular health score category. For both FMC and DC, our data provided little evidence for a true difference between the groups. Distribution of the data is displayed as violin plots (Figure 3).

## 4. Discussion

In this cross-sectional analysis that included participants aged 50–91 years and free of any cardiovascular disease, we found that ideal cardiovascular health was associated with lower cIMT, cLD, and baPWV. We found no differences for FMD or DC.

While we found statistically significant differences between intermediate and ideal cardiovascular health, these results beg the question of whether they are clinically relevant. For cIMT, Lorenz et al. (2007) performed a meta-analysis comprising eight studies and over 37,000 adults [37]. Adjusted for sex and age, a 0.1 mm difference in cIMT equated to an increased risk of myocardial infarction of 10% to 15%. Similarly, the risk of stroke increased by 13% to 18% per 0.1 mm cIMT difference. Given that the cIMT difference between the intermediate and ideal cardiovascular health groups was considerably lower in our study, the benefits of ideal over intermediate cardiovascular health metrics in an otherwise very healthy population remain controversial. This is in line with recent results of the meta-analysis by Fritze et al. (2020) [18]. The authors concluded that cIMT provided little information on cardiovascular risk and non-cardiovascular mortality. However, they found that higher cLD was associated with a higher hazard ratio for coronary heart disease and cardiovascular disease. A 1 mm difference in cLD equated to an increased cardiovascular risk of 29%. As with cIMT, however, the difference in cLD in our population was much lower than what can be considered clinically relevant.

As for baPWV, the meta-analysis by Vlachopoulos et al. (2012) showed that per 1 m/s increase, the risk of cardiovascular events, cardiovascular mortality, and all-cause mortality increased by 12%, 13%, and 6%, respectively. The baPWV difference found between the two groups in our study is considerably lower, again leaving the matter of its clinical relevance open to discussion.

Taken together, only three of the biomarkers of arterial structure and function showed statistically significant differences between the intermediate and ideal cardiovascular health groups in our study. Although these effects are relatively small, it seems noteworthy that ideal lifestyle-based cardiovascular health metrics still impact vascular aging, even in such a healthy population. In accordance with studies on people with a broader range of cardiovascular risk and a higher proportion of individuals with poor cardiovascular health [12,13,14], these results demonstrate the importance of promoting the highest possible adherence to Life’s Simple 7 in any population. However, reaching an intermediate cardiovascular health metric might be sufficient for healthy vascular aging in very healthy people with high fitness and low cardiovascular risk. This is in agreement with a recent meta-analysis that showed that, although meeting 5–7 of the Life’s Simple 7 metrics offered the highest protection from cardiovascular events, meeting 3–4 metrics was already associated with a 47% risk reduction when compared to those with a score of 0–2 [38]. With this study, we consolidate the case for using Life’s Simple 7 to assess cardiovascular health and make lifestyle recommendations. We demonstrated that in the healthiest part of the population aged 50+ years, meeting some health metrics was associated with similar vascular health to meeting most or all metrics. In contrast, other assessment tools, such as the Framingham Risk Score [39], heavily depend on non-modifiable metrics, such as age, sex, and race [40]. It is also well known that ultrasound-based cardiovascular risk predictors such as cIMT or FMD are profoundly influenced by those non-modifiable metrics. For example, recent findings in a healthy elderly population showed that 28.2% of cIMT variance in men and 23.9% in women was explained by age [41]. Similarly, using data from the same population as in the present study, we found that FMD decreased by 63.6% in men and 47.1% in women between the ages of 20 and 91 years [20]. Our results may, therefore, be of particular interest to, and motivating for, the general population, as Life’s Simple 7 has the advantage of being more comprehensive and communicative than other risk scores and solely consists of modifiable metrics.

### Strengths and Limitations

This study population consisted of middle-aged and older subjects with no manifested cardiovascular disease or history of cardiovascular events. Exclusion criteria included cardiovascular risk factors, such as obesity, smoking, and high blood pressure. Therefore, our study population was unique, representing a healthy population sample. Further, we used state-of-the-art measuring techniques for arterial properties, physical activity, and blood analyses and strictly adhered to current guidelines. This study, however, has several limitations. As with all cross-sectional approaches, results must be interpreted accordingly, as causation cannot be determined. Second, the strength of having a population free of cardiovascular disease and with only very few cardiovascular risk factors can also be seen as a limitation, as our results cannot be translated to the general population. Third, the diet assessment was carried out using a questionnaire that did not allow to conclude on all dietary components used in the Life’s Simple 7 score. Lastly, due to some missing data, we had to impute values, potentially leading to an overestimation of test statistics [42].

## 5. Conclusions

This study demonstrates that differences in arterial properties can be found even in a healthy population with either an intermediate or ideal cardiovascular health score. When maintaining a healthy lifestyle becomes more and more challenging, Life’s Simple 7 offers a comprehensive and communicable approach to improving and preserving vascular health over the life course. Ultrasound-based measurements of arterial properties, such as FMD, cIMT, and baPWV, are feasible and highly valuable tools to monitor the effects of lifestyle modifications in general, especially cardiovascular morbidity and mortality.

## Figures and Tables

**Figure 1 nutrients-14-03616-f001:**
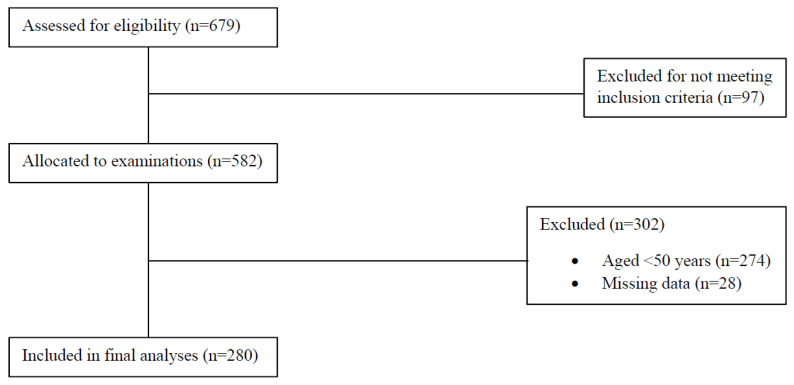
Flowchart of participant recruitment and data acquisition.

**Figure 2 nutrients-14-03616-f002:**
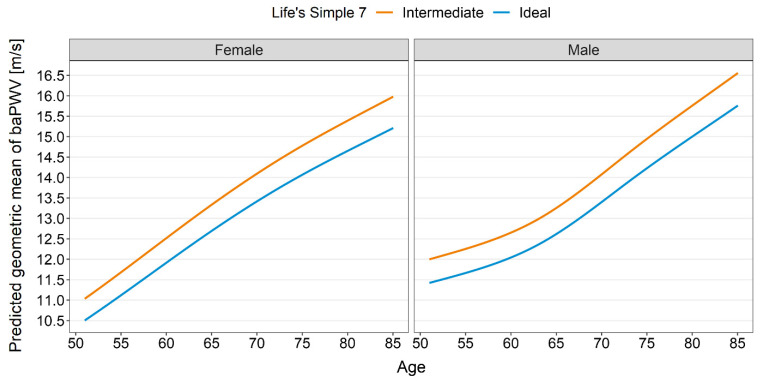
baPWV prediction by cardiovascular health score category.

**Figure 3 nutrients-14-03616-f003:**
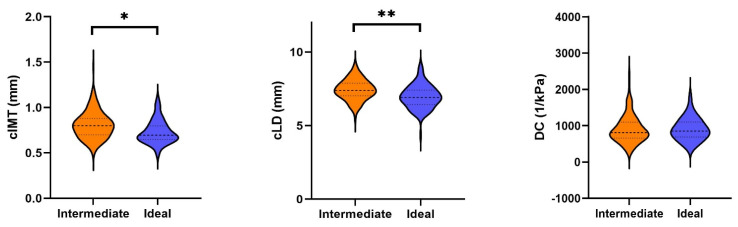
Violin plots of the arterial properties, divided into intermediate and ideal health scores. Legend: baPWV, pulse wave velocity; cIMT, carotid intima–media thickness; cLD, carotid lumen diameter; DC, distensibility coefficient; FMD, flow-mediated dilation. * indicates *p* < 0.05, ** indicates *p* < 0.01.

**Table 1 nutrients-14-03616-t001:** Participants’ characteristics, by cardiovascular health score category. *p*-value indicates the difference between the ideal and intermediate health scores.

	N	All	N	Ideal HS	N	Intermediate HS	*p*-Value
		Mean (SD)		Mean (SD)		Mean (SD)	
Male, n (%)	280	142 (50.7)	125	63 (50.4)	155	79 (51.0)	
Age	280	68.1 (10.7)	125	64.7 (10.1)	155	70.9 (10.4)	<0.001
cIMT (mm)	277	0.772 (0.144)	124	0.730 (0.127)	153	0.807 (0.148)	<0.001
cLD (mm)	216	7.21 (0.77)	103	6.96 (0.78)	113	7.43 (0.68)	<0.001
DC (l/kPa)	216	905.5 (344.9)	103	903.3 (308.0)	113	907.6 (376.7)	0.872
baPWV (m/s)	280	13.65 (2.49)	125	12.77 (2.22)	155	14.36 (2.48)	<0.001
FMD (%)	212	5.86 (3.66)	98	6.27 (3.69)	114	5.51 (3.62)	0.122
Systolic BP (mmHg)	280	131 (13)	125	125 (12)	155	136 (12)	<0.001
Diastolic BP (mmHg)	280	81 (8)	125	78 (8)	155	83 (8)	<0.001
Body mass index (kg/m2)	280	24.0 (2.7)	125	23.3 (2.4)	155	24.5 (2.8)	<0.001
Waist to hip ratio	280	0.90 (0.08)	125	0.88 (0.07)	155	0.91 (0.08)	<0.001
Low PA (min)	272	107 (34)	125	109 (36)	147	104 (32)	0.213
Moderate PA (min)	272	154 (65)	125	177 (62)	147	134 (60)	<0.001
Vigorous PA (min)	272	6 (9)	125	9 (11)	147	4 (7)	<0.001
Triglycerides (mg/dl)	278	121 (59)	124	126 (55)	154	125 (54)	0.119
Total cholesterol (mg/dl)	278	238 (38)	124	226 (34)	154	248 (39)	<0.001
Fruits and vegetables (portions/d)	277	3.5 (1.7)	124	4.2 (1.8)	153	3.0 (1.4)	0.024
Antihypertensives n (%)	280	49 (17.5)	125	11 (8.8)	155	38 (24.5)	<0.001
Beta-blocker n (%)	280	9 (3.2)	125	2 (1.6)	155	7 (4.5)	0.150

Legend: baPWV, brachial–ankle pulse wave velocity; BP, blood pressure; cIMT, carotid intima–media thickness; cLD, carotid lumen diameter; DC, distensibility coefficient; HS, Life’s Simple 7 cardiovascular health score; PA, physical activity. Antihypertensives include angiotensin-converting enzyme inhibitors, angiotensin receptor blockers, beta-blockers, and anticoagulants.

**Table 2 nutrients-14-03616-t002:** Estimated mean difference or ratio of geometric means of the arterial properties between ideal and intermediate cardiovascular health score. Based on multiple imputation models. Adjusted for age and sex.

Outcome Variable	Health Score	Estimate	95% Confidence Intervals		*p*-Value
cIMT (mm)	Ideal-Intermediate	−0.038	−0.069	−0.007	0.017
cLD (mm)	Ideal–Intermediate	−0.28	−0.46	−0.11	0.002
DC (1/kPa) ^1^	Ideal–Intermediate	0.94	0.85	1.05	0.207
FMD (%)	Ideal–Intermediate	0.31	−0.58	1.20	0.495
baPWV (%) ^1^	Ideal-Intermediate	0.95	0.92	0.98	0.003

Legend: baPWV, pulse wave velocity; cIMT, carotid intima–media thickness; cLD, carotid lumen diameter; DC, distensibility coefficient; FMD, flow-mediated dilation. ^1^ indicates geometrical mean ratios.

## Data Availability

The raw data supporting the conclusions of this article will be made available by the authors, without undue reservation.

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
