# Peer review of "Ideal Life’s Simple 7 Score Relates to Macrovascular Structure and Function in the Healthy Population"

_nutrients, 2022, doi:10.3390/nu14173616_

Round 1
Reviewer 1 Report
The manuscript under the title “Ideal Life’s Simple 7 score relates to macrovascular structure and function in the healthy population” represents an interesting original scientific paper describing the association of the Life’s Simple 7 cardiovascular health scores (from the American Heart Association with the macrovascular structure and function [Carotid intima-media thickness (cIMT), carotid lumen diameter (cLD), carotid distensibility coefficient (DC), flow-mediated dilation (FMD), and brachial-ankle pulse wave velocity (baPWV)] in a healthy sample of middle-aged and older adults (50 and 91 years of age) who were part of the COmPLETE Health study at the University of Basel (tested between January 2018 and June 2019). The study demonstrates that differences in arterial properties can be found even in a healthy population with either an intermediate or ideal cardiovascular health score.
Most of the manuscript is well written, with sufficiently explained Materials and method section. However, there are some mistakes presented in the Result section that have to be corrected before the manuscript is acceptable for publication.
These are:
The figure numbers have to be corrected (two figures were designated as figure 1: Flow-chart of participant recruitment and data acquisition, and baPWV prediction by cardiovascular health score category).
The numerical data for the frequency of participants associated with low, moderate, and vigorous PA should be corrected( 125 + 155 = 280, not 272).
The p values for the estimated difference for cLD (mm) and DC (1/kPa) are different in table 2 than in the manuscript text.
Also, it is not necessary to extensively repeat in the manuscript the numerical values already presented in table 2.
Figure 2 was not mentioned and explained in the manuscript text.
The authors have written: “Figure s (see supplements) shows the absolute means of baPWV, by cardiovascular health score category”- it should be more clearly explained where those data are presented.
The minor (-to-major- numerical data must be scrutinized and corrected) revision of the manuscript is recommended.
Author Response
Ideal Life’s Simple 7 score relates to macrovascular structure and function in the healthy population
Point-by-point answers
Reviewer 1:
The manuscript under the title “Ideal Life’s Simple 7 score relates to macrovascular structure and function in the healthy population” represents an interesting original scientific paper describing the association of the Life’s Simple 7 cardiovascular health scores (from the American Heart Association with the macrovascular structure and function [Carotid intima-media thickness (cIMT), carotid lumen diameter (cLD), carotid distensibility coefficient (DC), flow-mediated dilation (FMD), and brachial-ankle pulse wave velocity (baPWV)] in a healthy sample of middle-aged and older adults (50 and 91 years of age) who were part of the COmPLETE Health study at the University of Basel (tested between January 2018 and June 2019). The study demonstrates that differences in arterial properties can be found even in a healthy population with either an intermediate or ideal cardiovascular health score.
Most of the manuscript is well written, with sufficiently explained Materials and method section. However, there are some mistakes presented in the Result section that have to be corrected before the manuscript is acceptable for publication.
These are:
The figure numbers have to be corrected (two figures were designated as figure 1: Flow-chart of participant recruitment and data acquisition, and baPWV prediction by cardiovascular health score category).
The figure labelled “baPWV prediction by cardiovascular health score category.”, was submitted as figure s in the supplements (see results section, page 5). As it is now in the main text, we changed the figure to “figure 2”, and changed the following sentence in chapter 3, results, on page 5: Figure 2 shows the absolute means of baPWV, by cardiovascular health score category.
The numerical data for the frequency of participants associated with low, moderate, and vigorous PA should be corrected (125 + 155 = 280, not 272).
We corrected the number of participants. There were 147 participants with an intermediate health score, vs. 125 with an ideal health score. Changes are visible in table 1, page 6.
The p values for the estimated difference for cLD (mm) and DC (1/kPa) are different in table 2 than in the manuscript text.
We corrected the p-values for cLD and DC in table 2. The ones in the text we correct.
Also, it is not necessary to extensively repeat in the manuscript the numerical values already presented in table 2.
We shortened the section in the main text, describing only the data with significant differences between the groups.
Figure 2 was not mentioned and explained in the manuscript text.
We added the following sentence in the results section (page 6) and renamed the figure to figure 3: Distribution of the data are displayed as violin plots (figure 3).
The authors have written: “Figure s (see supplements) shows the absolute means of baPWV, by cardiovascular health score category”- it should be more clearly explained where those data are presented.
The publishers placed the figure in the main text. As the figure is now placed right after the section where it is described, we did not make any changes to the text. We opted for a visual presentation of the difference of baPWV, as not every reader may be familiar with geometric mean differences.
The minor (-to-major- numerical data must be scrutinized and corrected) revision of the manuscript is recommended.

Reviewer 2 Report
The data presented in the article Gilles Nève et al. they are an interesting addition to the knowledge about the positive impact of a healthy lifestyle on cardiovascular health, including the condition and properties of the arteries.
The manuscript submitted for review is clear, presented in a well-structured form, and is scientifically sound. The research methods are described in great detail and clearly. The data obtained in the study are clearly reflected in figures and tables. The interpretation of the data obtained is presented clearly and consistently throughout the entire text of the manuscript.
Statistical analysis of the data is carried out at the modern level, justified and understandable.
The cited references need to be updated, since only half (22 out of 42) of them are recent (over the last 5 years).
And, although in paragraph 4.1 the authors describe the strengths and weaknesses of the study, I have a number of questions:
1. I think it is not correct to call the sample under study healthy. Especially considering the inclusion criterion, such as blood pressure <160/100 mmHg.
According to ESC/ESH, blood pressure <130/85 is considered normal, high normal (HPA) — 130- 139/85-89 , Grade 1 AG 140-159/90-99, Grade 2 AG 160-179/100-109, Grade 3 AG ≥180/110 mmHg. Please make changes.
2. There is also no report of taking antihypertensive drugs by the participants of the study group.
3. Section 2.1 states that in order to be included in the COmPLETE Health Study, all participants had to have been non–smokers for at least 10 years, i.e., former smokers. At the same time, in the full protocol - reference [24] – the past status of a smoker is a criterion for exclusion from the study.
4. The research hypothesis is not clearly indicated. Please complete it.
In general, the results presented in the manuscript are undoubtedly of interest to a professional audience regarding the use of this type of monitoring of risk factors for cardiovascular diseases, as well as in the development of effective measures to maintain a healthy lifestyle.
Author Response
Ideal Life’s Simple 7 score relates to macrovascular structure and function in the healthy population
Point-by-point answers
Reviewer 2
The data presented in the article Gilles Nève et al. they are an interesting addition to the knowledge about the positive impact of a healthy lifestyle on cardiovascular health, including the condition and properties of the arteries.
The manuscript submitted for review is clear, presented in a well-structured form, and is scientifically sound. The research methods are described in great detail and clearly. The data obtained in the study are clearly reflected in figures and tables. The interpretation of the data obtained is presented clearly and consistently throughout the entire text of the manuscript.
Statistical analysis of the data is carried out at the modern level, justified and understandable.
The cited references need to be updated, since only half (22 out of 42) of them are recent (over the last 5 years).
We believe that our references are well chosen, as we cite the most impactful and largest studies, not necessarily the most recent ones. In addition, some of the references such as the early ones on the Life’s Simple 7 or the validation of certain methods used in our manuscript, date from the early- to mid-2010’s.
And, although in paragraph 4.1 the authors describe the strengths and weaknesses of the study, I have a number of questions:
- I think it is not correct to call the sample under study healthy. Especially considering the inclusion criterion, such as blood pressure <160/100 mmHg.
According to ESC/ESH, blood pressure <130/85 is considered normal, high normal (HPA) — 130- 139/85-89 , Grade 1 AG 140-159/90-99, Grade 2 AG 160-179/100-109, Grade 3 AG ≥180/110 mmHg. Please make changes.
We think that calling the population “healthy” is accurate. Mild arterial hypertension is considered a cardiovascular risk factor but none of our participants had any cardiovascular disease. We set the inclusion criteria to <160/100 mmHg to firstly get a representative sample of the disease-free population (including those with cardiovascular risk factors), and secondly, not having to exclude too many participants due to “white scrubs effect” because of blood pressure measurements in an unknown and potentially stressful environment.
- There is also no report of taking antihypertensive drugs by the participants of the study group.
We added, in table 1 on page 6, the use of antihypertensives and beta-blockers.
- Section 2.1 states that in order to be included in the COmPLETE Health Study, all participants had to have been non–smokers for at least 10 years, i.e., former smokers. At the same time, in the full protocol - reference [24] – the past status of a smoker is a criterion for exclusion from the study.
We hereby refer to the study by Jatoi et al., 2007 (https://doi.org/10.1161/HYPERTENSIONAHA.107.087338), who show the reversibility of arterial stiffness after 10 years of smoking cessation. We therefore consider those who have stopped tobacco smoking for >10 years to be non-smoker equivalents.
- The research hypothesis is not clearly indicated. Please complete it.
We apologize that the introduction was missing in your first review of the manuscript. In it, we clearly state the aim of this article on page 2: In the general population, the Life’s Simple 7, and various ultrasound-based assessment methods of arterial structure and function are independently known to be influenced by age and lifestyle. Both have been used separately to assess cardiovascular health but evidence of their association is scarce, especially in healthy, middle-aged to elderly populations. Therefore, this study aims to assess the association between the Life’s Simple 7 cardiovascular health score and clinical biomarkers of structure and function of the main arteries in healthy community-dwelling participants.
In general, the results presented in the manuscript are undoubtedly of interest to a professional audience regarding the use of this type of monitoring of risk factors for cardiovascular diseases, as well as in the development of effective measures to maintain a healthy lifestyle.
